# Chronic hepatitis B in remote, tropical Australia; successes and challenges

Josh Hanson [1,2]*, Melissa Fox[2], Adam Anderson[2], Penny Fox[2], Kate Webster[2], Charlie Williams[3], Blake Nield[4], Richard Bagshaw[2], Allison Hempenstall[5], Simon Smith[2], Norma Solomon[2], Peter Boyd[2]

1 The Director's Unit, The Kirby Institute, University of New South Wales, Sydney, New South Wales, Australia, 2 Division of Medicine, Cairns Hospital, Cairns, Queensland, Australia, 3 Department of Medicine, Royal Darwin Hospital, Darwin, Northern Territory, Australia, 4 Department of Microbiology, St George Hospital, Sydney, New South Wales, Australia, 5 Thursday Island Hospital, Thursday Island, Queensland, Australia

* jhanson@kirby.unsw.edu.au

**Data Availability Statement:** Data cannot be shared publicly because of the Queensland Public Health Act 2005. Data are available from the Far North Queensland Human Research Ethics Committee (contact via Cairns_Ethics@health.qld.

## Abstract

### Introduction

Aboriginal and Torres Strait Islander Australians living in remote locations suffer disproportionately from chronic hepatitis B (CHB). Defining the temporospatial epidemiology of the disease—and assessing the ability of local clinicians to deliver optimal care—is crucial to improving patient outcomes in these settings.

### Methods

The demographic, laboratory and radiology findings in all patients diagnosed with CHB after 1990, and presently residing in remote Far North Queensland (FNQ), tropical Australia, were correlated with their management and clinical course.

### Results

Of the 602 patients, 514 (85%) identified as Aboriginal and Torres Strait Islander Australians, 417 (69%) of whom had Torres Strait Islander heritage. Among the 514 Aboriginal and Torres Strait Islander Australians, there were only 61 (12%) born after universal postnatal vaccination was introduced in 1985. Community CHB prevalence varied significantly across the region from 7/1707 (0.4%) in western Cape York to 55/806 (6.8%) in the Eastern Torres Strait Islands. Although 240/602 (40%) are engaged in care, with 65 (27%) meeting criteria for antiviral therapy, only 43 (66%) were receiving this treatment. Among 537 with complete data, 32 (6%) were cirrhotic, of whom 15 (47%) were engaged in care and 10 (33%) were receiving antiviral therapy. Only 64/251 (26%) in whom national guidelines would recommend hepatocellular carcinoma (HCC) surveillance are receiving screening, however, only 20 patients have been diagnosed with HCC since 1999.

### Conclusion

Vaccination has had a dramatic effect on CHB prevalence in FNQ in only a generation. However, although engagement in care is the highest in Australia, this is not translating into

gov.au) for researchers who meet the criteria for access to confidential data.

**Funding:** The authors received no specific funding for this work.

**Competing interests:** The authors have declared that no competing interests exist.

initiation of antiviral therapy in all those that should be receiving it, increasing their risk of developing cirrhosis and HCC. New strategies are necessary to improve the care of Indigenous Australians living with CHB to reduce the morbidity and mortality of this preventable disease.

## Introduction

The prevalence of chronic hepatitis B (CHB) is low in Australia, compared with other countries in the Asia-Pacific region, but its Indigenous Aboriginal and Torres Strait Islander people (hereafter respectively referred to as Indigenous Australians) bear a disproportionate burden of the disease [1]. At the end of 2017, the prevalence was 4.0% among Indigenous Australians compared with a prevalence of 0.95% in the general population [1, 2]. The higher prevalence of CHB contributes to the higher rates of liver disease seen in Indigenous Australians, particularly liver failure and hepatocellular carcinoma (HCC). Nationally, liver-related deaths are 3.7 times more common in Indigenous than in non-Indigenous Australians [3].

Guidelines have been published to improve the care of CHB in Australia [4, 5]. These guidelines emphasize the importance of education and retention in care, the prescription (where appropriate) of anti-viral therapy, the treatment of comorbidities that increase the risk of liver failure and HCC, and active screening for these complications. However, there are significant challenges in delivering this care to the many Indigenous Australians who live in remote locations, where access to health care is limited and where other health conditions compete for finite resources [6, 7].

Most of the work examining the burden of CHB in Indigenous people in remote Australia has been performed in the Northern Territory of Australia. Local clinicians have produced an impressive body of work describing the epidemiology of the infection, particularly the ubiquity of the C4 viral genotype and the burden of HCC between 1990 and 2010 [8–10]. However, the prospective generalisability of these findings to Indigenous Australians living in other parts of remote Australia has not been defined. These Northern Territory data have led to recommendations for HCC screening with ultrasound and alpha-fetoprotein every six months in all Indigenous patients with CHB over the age of 50 years [5], however, the incremental benefit of this strategy above appropriate anti-viral prescription and improved management of comorbidities is uncertain.

The Cape York Peninsula and Torres Straits Islands, an area of almost 160,000km$^2$ in remote Far North Queensland (FNQ) in tropical Australia, has a resident population of 26,514, almost 69% of whom identify as Aboriginal and/or Torres Strait Islander people [11]. The published estimated local prevalence of CHB of 3.12% is one of the highest in Australia [2], although there is likely to be a variation in the burden of disease across the 35 separate communities that are dispersed widely across the region. There are also likely to be significant disparities between the burden of disease in the Aboriginal and Torres Strait Islander people, ethnologically and genetically distinct populations with very different histories.

A vaccination programme began in new-born Indigenous infants in the region in 1985 at a time when there were several Indigenous communities in FNQ where greater than 20% of the tested children were hepatitis B surface antigen (HBsAg) positive [12]. However, the impact of this programme—and the implications for current disease management strategies—has not been evaluated. Anecdotally clinicians in FNQ do not see the burden of liver related disease to CHB that has been described in the Northern Territory. Whether this is truly the case or simply an under-recognition of the scale of the problem remains to be determined.

This study was performed to evaluate the temporospatial epidemiology of the disease in remote FNQ, to define the present burden of HBV infection. The study also assessed local clinicians' adherence to components of HBV care that are currently recommended in National Guidelines [4, 5]. It was hoped that these data might identify the successes of current locally employed strategies and the challenges that remain.

## Methods

This retrospective audit was performed using data collected from the Queensland Notifiable Conditions System (NOCS) database. Hepatitis B (HBV) is a notifiable infection in the state of Queensland: all patients without prior evidence of HBV infection who test positive for HBsAg or hepatitis B virus deoxyribose nucleic acid (HBV DNA) are reported to the state's HBV disease register. The study period of January 1, 1990 to December 31, 2019 was chosen as this coincided with the completion of the initial phase of the Queensland Hepatitis B program [12]. Additional data were collected from the FNQ HBV clinical database, which is populated by the NOCS database and which also draws data from local public and private laboratory and radiology services to improve the delivery of patient care.

Living residents of FNQ with confirmed CHB (HBsAg positive on two occasions, 6 months apart) in the FNQ HBV database were included in the study. Demographic, clinical, laboratory and radiological data were collected from the database and patient medical records. Patients were defined as being "engaged in care" if they were receiving anti-viral therapy or had had a HBV DNA test performed in the preceding 12 months [2]. Patients were defined as being eligible for treatment if they were in the immune clearance, immune escape phases or had cirrhosis, as per Australian national guidelines [4]. Cirrhosis was said to be present if it was identified on liver biopsy or imaging (ultrasound, computed tomography or magnetic resonance imaging) suggested cirrhosis or portal hypertension. Cirrhosis was also said to be present if there was an AST to Platelet Ratio Index (APRI) score of >2 [13], a transient elastography score of >11.7kPa [14] or a clinical diagnosis of cirrhosis by a physician.

Cases of HCC were identified using the Queensland Cancer Registry from 1999 to 2016 (the period for which data were available) and cross checked against medical records. Cases after 2016 were identified from the medical records of patients in the FNQ HBV database. Data were entered into an electronic database (Excel; Microsoft, Redmond, WA, USA), aggregated and presented in a descriptive manner.

The study was approved by the Far North Queensland Human Research Ethics Committee (QCH106-1082 and RD006103). As the study was retrospective and the data were to be presented in a de-identified and aggregated manner, the committee waived the requirement for informed consent. However, individual communities were not identified to prevent the possibility of stigmatisation.

## Results

There were 1429 cases of CHB in recorded on the NOCS database in remote FNQ during the study period. After review of longitudinal demographic, clinical and laboratory data, 827 cases were excluded (Fig 1) leaving 602 people living with CHB in remote FNQ, representing 2.27% of the region's general population of 26,514.

The median (interquartile range (IQR)) age of the 602 patients was 46 (38–58) years, 322 (53.5%) were male. Of the 602 patients, 514 (85.4%) identified as Indigenous Australians, this included 374 (62.1%) who identified as Torres Strait Islander, 97 (16.1%) who identified as Aboriginal and 43 (7.1%) who identified as both. Among the remaining 88 patients, the

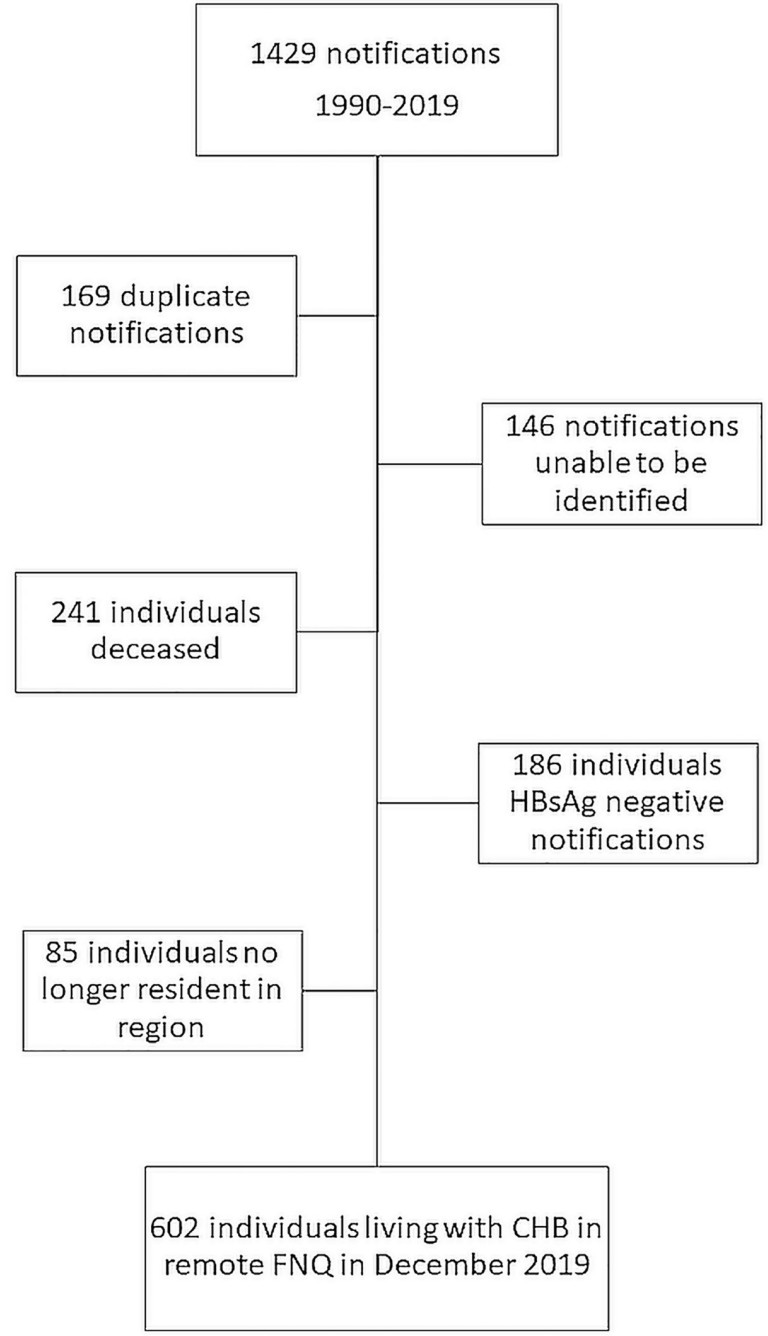

**Fig 1. Flow diagram showing identification of patients currently living with CHB in remote FNQ.** Patients were excluded if they were duplicate NOCS notifications due to name spelling errors or variations of date of birth, had subsequently cleared the virus (HbsAg negative on two occasions), had died or no longer resided in FNQ, or were unable to be identified despite contacting the clinic where the patients were initially diagnosed and last known to have had blood tests.

country of birth was recorded in 73 with 42 (57.5%) born overseas (predominantly in the Asia-Pacific region, most commonly Papua New Guinea) and 31 (42.5%) being born in Australia (S1 Table). Among the 545 patients born in Australia, 62 (11.4%) were born after 1985, 61 (98.4%) of whom were Indigenous Australians (Fig 2).

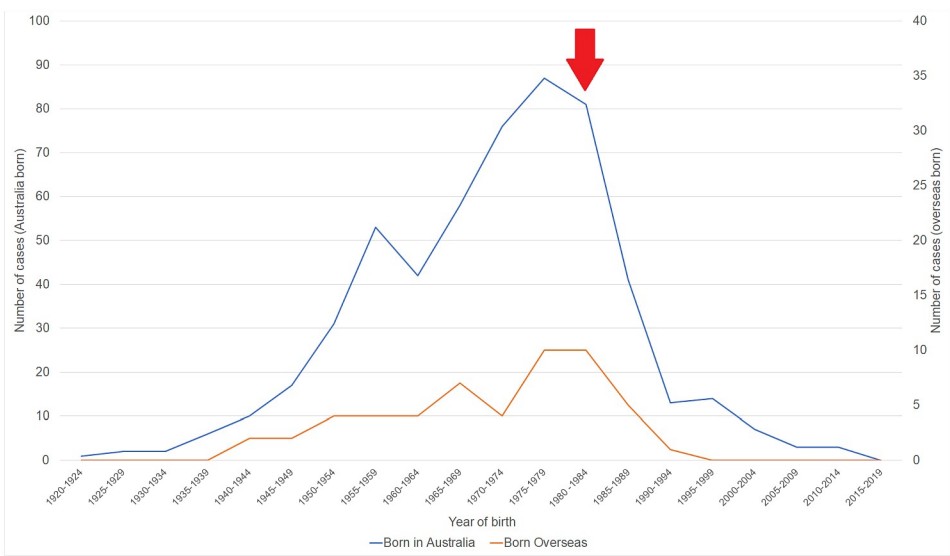

**Fig 2. The number of individuals living in rural FNQ living with CHB stratified by date of birth and country of birth.** The arrow highlights the year (1985) when universal post-natal vaccination began in Indigenous communities in FNQ, demonstrating the effect of vaccination on local incidence.

The prevalence in different communities varied significantly. At the end of the study period, the community prevalence in some island clusters in the Torres Straits was greater than 6%, while in several Aboriginal communities on Cape York the community prevalence was less than 1% (Fig 3).

Of the 602 patients, 240 (39.9%) were engaged in care, 65 (27.1%) of whom met criteria for anti-viral therapy, but only 43 (66.2%) were receiving it. Among the 537/602 (89.2%) who had sufficient data to determine its presence, 32 (6.0%) had cirrhosis, 15 (46.9%) of whom were engaged in care, 10 (31.3%) of whom were receiving therapy. Of the 251 patients meeting national guidelines for screening, 64 (25.5%) had had an ultrasound in the prior 12 months. The proportion of patients engaged in care and receiving therapy was similar across the region (Table 1).

Between 1999–2016 there were 41 cases of HCC in residents of remote FNQ notified to the Queensland cancer registry, 36 (88%) occurred in Indigenous Australians. Of the 41 cases, 40 were tested for HBV, 18 (45%) were HBsAg positive, an additional 2 cases were hepatitis B core antibody (HBcAb) positive, but HBsAg negative. Two additional cases were identified in the 2016–2019 period by reviewing the FNQ clinical database; there were no cases identified in the FNQ database that had not been reported to the Queensland cancer registry. The median (IQR, range) age at diagnosis among the 20 HBsAg positive HCC cases was 57 (60–65, 41–77) years, 15/20 (75%) were male and 19 (95%) were Torres Strait Islander individuals; there was not a single case in an Aboriginal Australian during the study period. Among the 20 HCC patients, 11/13 (85%) with complete data had cirrhosis, while only 2/11 (18%) with complete data were receiving anti-viral therapy. Of the 16 patients with treatment documentation available, 13 (65%) received palliative care, 1 (6%) received surgery and 2 (13%) received chemotherapy. Among the 20 patients, 19 (95%) had died after a median (IQR, range) of 88 (46–221, 10–788) days. The single patient in whom death was not confirmed had returned to his country of birth and lost to follow-up.

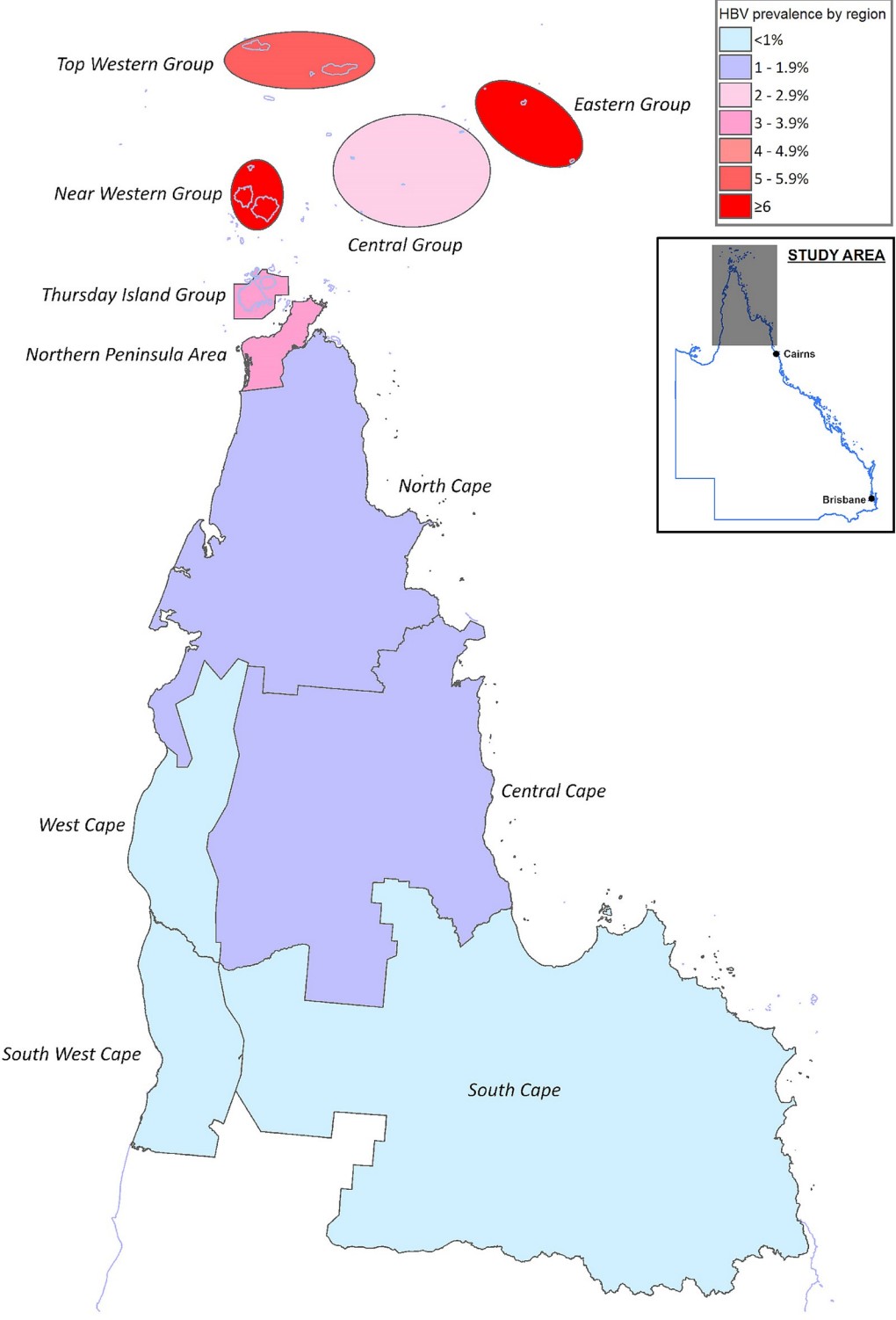

**Fig 3. CHB prevalence across the region.** The map was created using constructed using mapping software (MapInfo version 15.02, Connecticut, USA) using data provided by the State of Queensland (QSpatial). Queensland Place Names— State of Queensland (Department of Natural Resources, Mines and Energy) 2019, available under Creative Commons Attribution 4.0 International licence https://creativecommons.org/licenses/by/4.0/. 'Coastline and state border— Queensland—State of Queensland (Department of Natural Resources, Mines and Energy) 2019, available under Creative Commons Attribution 4.0 International licence https://creativecommons.org/licenses/by/4.0/.

**Table 1. Selected demographics, clinical characteristics, and engagement in care of individuals living in rural FNQ living with CHB stratified by remoteness.**

|  | South Cape York n = 45 | North Cape York n = 187 | Inner Torres Straits Islands n = 144 | Outer Torres Straits Islands n = 226 |
|---|---|---|---|---|
| Born in Australia | 33 (73%) | 182 (97%) | 126 (88%) | 204 (90%) |
| Born after universal vaccination | 6 (13%) | 23 (12%) | 17 (12%) | 21 (9%) |
| Indigenous Australian [a] | 31 (69%) | 169 (90%) | 112 (78%) | 202 (89%) |
| Torres Strait Islander Australian | 3 (7%) | 111 (59%) | 106 (74%) | 197 (87%) |
| Engaged in care [b] | 13 (29%) | 57 (30%) | 50 (35%) | 120 (53%) |
| Meet criteria for treatment [c] | 4/14 (29%) | 17/62 (27%) | 20/58 (34%) | 36/123 (29%) |
| Number on treatment | 2/4 (50%) | 10/17 (59%) | 9/20 (45%) | 22/36 (61%) |
| Cirrhosis [d] | 2/42 (5%) | 9/170 (5%) | 11/117 (9%) | 10/208 (5%) |
| Meet criteria for HCC screening [c] | 18 (40%) | 76 (41%) | 60 (42%) | 98 (43%) |
| Ultrasound in last 12 months | 3/18 (17%) | 19/76 (25%) | 10/60 (17%) | 32/98 (33%) |

[a] Identifies as an Aboriginal Australian, a Torres Strait Islander Australian or both.

[b] HBV DNA measured in 2019 or on therapy;

[c] as per Australian national guidelines [4]. Some of the patients who met criteria for therapy were not engaged in care (prior documentation of cirrhosis);

[d] of the 537 with sufficient data to determine this

## Discussion

Vaccination has had a dramatic effect on the prevalence of CHB in remote FNQ, although it is profoundly disheartening that children have been born with the disease in Australia in the 21st century. The local burden of CHB is, unsurprisingly, disproportionately borne by Indigenous Australians, although there is remarkable variation in disease prevalence across the region, with the Torres Strait Islander population particularly affected. While the number of patients engaged in care is higher than in any other part of Australia [2], many that meet criteria for anti-viral therapy are not receiving it, contributing to the local burden of cirrhosis and HCC. Despite this, HCC was not identified frequently, with only approximately one case identified in the region annually, raising questions about the cost-efficacy of HCC screening locally. It was notable that there was not a single HCC identified in a Aboriginal Australian during the study period, a finding quite at odds with reports from other parts of the country [8].

Universal vaccination of Indigenous newborns was introduced in Queensland in 1985. A catch-up programme was introduced for indigenous children aged up to 10 years in 1987 and up to 18 years in 1989. When this programme commenced there were several communities in the Torres Strait Islands and on the Cape York peninsula where over 20% of the children were HBsAg positive [12]. The CHB prevalence has fallen impressively across the region in a single generation, although it is dispiriting that there have been 62 children born since 1985 who now have CHB, 61 of whom are Indigenous. Evidently, managing the vaccine cold chain across a tropical expanse of almost 160,000km$^2$ is challenging and, in the past, there was at least one large community where children missed out on the catch-up program [15]. Acknowledging these past failures, immunization rates have been improving: in 2017, 91.5% of Indigenous infants in FNQ completing HBV vaccination by 12 months [16]. There are also now expanded efforts to identify women of child-bearing age prior to conception to ensure optimal antenatal and peripartum care, and indeed no child born after 2014 has been identified to be HbsAg positive (S1 File). It would be anticipated that strategies that improve vaccination rates and optimise perinatal care will further reduce the prevalence of HBV in the future.

However, a significant proportion of the local Indigenous population—born before the advent of the HBV vaccine—continue to live with CHB. It is notable that there is marked

variation in the prevalence of CHB varies across the region. Communities in the south and west of Cape York have a disease prevalence (0.49% and 0.41% respectively) which is lower than that of Australia's general population (0.95%) [2]. In contrast, the community prevalence remains greater than 6% on many islands in the Torres Straits, higher than that which is seen in some Southeast Asian countries [17]. It is important to consider this locoregional variation in prevalence when planning the distribution of health resources and the cultural appropriateness of community education programs [18].

Even in well-resourced metropolitan settings, the Australian health system struggles to deliver optimal care to those living with CHB. The most recent national data estimate that 36% of people living in Australia with CHB are undiagnosed, that only 20% are engaged in care, and that less than half of the 20% who are estimated to require treatment are currently receiving it [2]. These figures are all the more striking given that Australia is recognised as one of the world's leaders in delivering care to people living with human immunodeficiency virus (HIV) and hepatitis C virus (HCV) [19, 20]. This may be because HBV is, in many ways, a more complicated disease to manage than HIV and HCV. Not all patients require therapy, but if they do it may be lifelong; meanwhile predicting the longitudinal course of the infection is difficult [21]. Although there are presently more people living in Australia with CHB than with HIV or HCV [2, 19, 20], Australians living with CHB are more likely to be Indigenous and more likely to be living in remote locations, emphasising that despite Australia's well-resourced health system, significant iniquities still remain [6].

In these remote locations there are challenges in delivering optimal care. These include mobile populations, the high turnover of clinical staff, challenges in delivering culturally appropriate care to the predominantly Indigenous patients, competing health priorities and limited health system infrastructure, particularly access to ultrasound and FibroScan [7, 22–25]. However while these challenges certainly exist, it has been possible to overcome many of these challenges in managing HCV in remote FNQ, with local clinicians now struggling to find patients to treat [26]. Furthermore, analysis of the local cascade of care also suggests that there is little difference in the proportion of patients that are engaged in care across the region, even the outlying islands of the Torres Straits. Indeed the 53% of the patients on these islands engaged in care is higher than in any primary health network in Australia [2].

However, although engagement in care in remote FNQ is, by national standards, relatively high, the prescription of antiviral therapy could be improved. Almost 30% of the patients engaged in care at the end of the study met criteria for antiviral treatment, but only 66% of these patients were receiving it. Potential explanations include a period of observation during the immune clearance phase, patient concerns about polypharmacy or clinicians' concerns about adherence [27]. It is true that the rate of antiviral prescription in remote FNQ is higher than in most Australian jurisdictions—the average proportion of patients with CHB receiving therapy is 8.3% [2]. However, as the median age in this cohort is only 46, it is important to identify strategies to increase antiviral prescription and uptake now to reduce the burden of cirrhosis and HCC in the future [28]. To achieve this end it is essential to have adequately trained and supported Aboriginal and Torres Strait Islander workforce as they are more likely to have an insight into the personal, community and organisational factors that may influence engagement with care and long-term adherence to antiviral therapy [24, 29, 30].

Australian guidelines recommend screening with 6 monthly ultrasound and alpha fetoprotein in all people with cirrhosis and in high risk populations that include Aboriginal and Torres Strait Islander people over the age of 50 [5]. While over 40% of the cohort in this study satisfied criteria for HCC surveillance, only a quarter of those eligible had received an ultrasound in the prior 12 months. That being said, HCC surveillance remains a controversial area and even in well-resourced Australian settings, adherence to surveillance recommendations

are as low as 10% [31–34]. There have only been two randomized controlled trials to examine the issue of HCC screening in patients with CHB: the first—a methodologically flawed study performed in Shanghai, China—showed a mortality benefit of biannual ultrasound screening [35], while a study examining the use of AFP in Qidong, China showed no mortality benefit [36]. The applicability of these recommendations developed in metropolitan China to Indigenous people in remote Australia is uncertain.

Data that examined HCC in the Northern Territory identified 15 cases of HCC in HBsAg positive Aboriginal Australians between 2000 and 2011 [8]. Using estimates of a seroprevalence of 8% in Indigenous people over 40 and a contention that 60% of HCC were attributable to HBV, this translated into an individual annual risk of 0.34% for 50–59-year-olds to 0.86% for 70–79-year-olds, above a cut-off of 0.2% which in American guidelines made screening for HCC cost-effective [37]. This justified—in the authors' view—a call for HCC surveillance in all Indigenous Australians with CHB from the age of 50 years [8]. However, the number of patients receiving anti-viral therapy in this series was not reported, an important issue, as therapy may reduce HCC—in both cirrhotic and non-cirrhotic patients—by up to 75% [38, 39]. Would screening still be cost-effective in this situation when the incidence of HCC would be expected to be lower? Modelling studies suggested that focussing on the prescription of antiviral therapy is likely to be far more cost-effective [40].

There are other factors that influence HCC risk: over 80% of HCCs in HBV occur in patients with cirrhosis [13, 39], in this series it was present in 85%, in the Northern Territory series it was 57% [8]. Meanwhile men are affected three to four times more commonly than women [21], an observation that was once again repeated in this series. Other factors including family history of HCC, a high level of viral replication, the degree of transaminitis and comorbidities also contribute [32, 41, 42]. These data were not presented in the Northern Territory series [8]. The HBV genotype also influences HCC risk, with genotype C having the strongest potential, possibly because HBeAg seroconversion occurs later in patients with genotype C infection [43, 44]. The ubiquity of the novel C4 genotype has been described in the Northern Territory, indeed in one series from that region Aboriginal patients were exclusively infected with this strain [10]. The infecting genotypes in FNQ have not been as clearly defined, although 3 of 5 isolates from a 2001 study of Aboriginal Australians living in Queensland were Genotype D which is less oncogenic [44, 45]. Against this background, it is notable that in this series there was not a single case of HCC in an Aboriginal person in FNQ in the entire study period [46].

Targeted screening which uses both patient and viral factors to individualize risk have been proposed although it has been difficult to validate these in different populations and in different geographical regions [41, 42]. While more targeted screening may be desirable—and likely more cost-effective—this is not currently embraced with the existing Australian guidelines, that recommend universal ultrasound and AFP testing [4]. Prospective studies are underway that will hopefully resolve the issue.

Limitations of this study include its retrospective methodology, resulting in incomplete data, especially the documentation of comorbidities. Demographic data recorded in the State-wide register sometimes documents Indigenous status inaccurately. It is likely that some of the individuals born in Australia are incorrectly classified as non-Indigenous. It is almost certain that there are some patients living with CHB who have not yet been diagnosed; this may particularly be the case with children in whom hepatitis screening—and blood testing generally—is performed less frequently. A previous Australian study found that using a state-wide cancer registry to identify cases of HCC underestimated their number by 50% primarily due to under reporting of clinical diagnosis of HCC [47]. However, while cases are likely to have been missed, it seems less likely that such a large number would be missed in rural FNQ where

almost all patients with significant illness are referred for evaluation and this data would be captured in the local clinical database.

## Conclusions

This study demonstrates both the successes and the challenges of delivering optimal CHB in remote tropical Australia. The impact of vaccination has been dramatic and the engagement in care is the highest recorded in Australia, however, antiviral therapy is still not being prescribed to all the patients in whom it is indicated. The barriers to its prescription need identification and remedying to prevent this predominantly middle-aged Indigenous cohort developing cirrhosis and HCC in future decades. An expanded Aboriginal and Torres Strait islander workforce will be central to achieving these aims. Although adherence to recommended HCC surveillance guidelines is poor, clinically apparent HCC appears to be relatively uncommon. Targeting HCC surveillance to individuals at higher risk of the disease, is likely to be more cost-effective than universal screening in remote Australian populations, allowing redeployment of finite resources to higher value interventions.

## Supporting information

**S1 Table. Country of birth of individuals living in rural FNQ living with CHB.**
(DOCX)

**S1 File. Local recommendations for antenatal and perinatal care.**
(PDF)

## Acknowledgments

The authors would like to acknowledge all the health professionals involved in the care of patients living with hepatitis B in the region.

## Author Contributions

**Conceptualization:** Josh Hanson, Adam Anderson, Simon Smith.

**Data curation:** Josh Hanson, Melissa Fox, Adam Anderson, Penny Fox, Kate Webster, Blake Nield, Richard Bagshaw.

**Formal analysis:** Josh Hanson, Melissa Fox, Adam Anderson.

**Investigation:** Josh Hanson, Melissa Fox, Adam Anderson, Kate Webster, Charlie Williams, Allison Hempenstall, Peter Boyd.

**Methodology:** Josh Hanson.

**Project administration:** Penny Fox, Peter Boyd.

**Supervision:** Josh Hanson, Adam Anderson, Penny Fox, Simon Smith, Norma Solomon, Peter Boyd.

**Visualization:** Josh Hanson, Simon Smith.

**Writing – original draft:** Josh Hanson, Melissa Fox, Adam Anderson.

**Writing – review & editing:** Josh Hanson, Melissa Fox, Adam Anderson, Penny Fox, Kate Webster, Charlie Williams, Blake Nield, Richard Bagshaw, Allison Hempenstall, Simon Smith, Norma Solomon, Peter Boyd.

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
