## [Decision Letter · Decision Letter 0]

20 Jul 2020

PONE-D-20-17114

Optimising the care of patients with chronic hepatitis B in remote, tropical Australia

PLOS ONE

Dear Dr. Hanson,

Thank you for submitting your manuscript to PLOS ONE. After careful consideration, we feel that it has merit but does not fully meet PLOS ONE’s publication criteria as it currently stands. Therefore, we invite you to submit a revised version of the manuscript that addresses the points raised during the review process.

Your manuscript was reviewed by 2 experts in the field. They found many important problems in your submission and provided copious comments. Please carefully review the attached comments and respond point-by-point.

We look forward to receiving your revised manuscript.

Kind regards,

Yury E Khudyakov, PhD

Academic Editor

PLOS ONE

Journal Requirements:

2. To comply with PLOS ONE submission guidelines, in your Methods section, please provide additional information regarding your statistical analyses.

In addition, please report your p-values to support your claims.

For more information on PLOS ONE's expectations for statistical reporting, please see https://journals.plos.org/plosone/s/submission-guidelines.#loc-statistical-reporting

Reviewers' comments:

Reviewer's Responses to Questions

**Comments to the Author**

1. Is the manuscript technically sound, and do the data support the conclusions?

Reviewer #1: Partly

Reviewer #2: Yes

2. Has the statistical analysis been performed appropriately and rigorously? 

Reviewer #1: Yes

Reviewer #2: N/A

3. Have the authors made all data underlying the findings in their manuscript fully available?

Reviewer #1: No

Reviewer #2: No

4. Is the manuscript presented in an intelligible fashion and written in standard English?

Reviewer #1: Yes

Reviewer #2: Yes

5. Review Comments to the Author

Reviewer #1: This descriptive manuscript reports the prevalence of chronic hepatitis B in remote Australia amongst aboriginal and Torres Strait Islander Australians. They report that 12% of individuals identified with hepatitis B in this region had chronic hepatitis B after introduction of universal vaccination. Of those engaged in care two thirds who qualified for antiviral therapy were receiving appropriate antiviral therapy. Low rates of hepatocellular carcinoma screening were also reported.

The article is important in that it demonstrates an inequity in access to healthcare within the robust Australian system. However the manuscript is quite lengthy, for the amount of data presented, particularly the discussion.

I would suggest the following modifications

1, The discussion is quite long for the data presented. The solutions proposed by the authors certainly deserve further attention but many of the suggestions are not really related to hepatitis B but rather related to improving general overall health (smoking etc.). The authors truncate the discussion some. As a minor point, I might change the title of the manuscript to something like”Epidemiology of hepatitis be in remote tropical Australia challenges and opportunities” or something similar.

2. There are a fair number of individuals with hepatitis B were born after universal vaccination. A description of the prenatal care available in this region of Australia would be very helpful for the readers to gain better context of why infants may not be vaccinated.

3. To place the shortcomings of care of hepatitis B patients in this remote part of Australia, can the authors compare rates of liver cancer, appropriate initiation of antiviral therapy, and hepatocellular cancer screening from this region to other more developed parts of Australia.

4. The authors note that hepatitis C screening has been quite successful in this region. Australia had a robust effort to eliminate hepatitis C. Are the same public health efforts being made for hepatitis B?

5. While the HCC screening rates are disappointing, the authors should note that their screening rates are not different than other parts of the world. Can the authors also compared to more developed regions of Australia.

Reviewer #2: In Australia, prevalence among its Indigenous Aboriginal and Torres Strait Islander people (Indigenous Australians) is disproportionately high, at 4.0% compared to 0.95% in the general population. Subsequently liver-related deaths in Australia re 3.7 times more common in Indigenous than in non-Indigenous Australians. Many Indigenous Australians live in remote

locations, where access to health care is limited and where other health conditions compete for

finite resources. This study evaluates the temporospatial epidemiology of HBV infection and clinical burden of disease in remote Far North Queensland (where the majority of the population identify as Aboriginal and/or Torres Strait Islander people) after the introduction of the hepatitis B vaccine. It was hoped that these data might define the scale of the local problem and inform future cost effective strategies to reduce CHB-related morbidity and mortality in the region.

This a retrospective study using data from the hepatitis B disease registry in the Queensland notifiable conditions system (NOCS) database, from January 1, 1990 to December 31, 2019 (coinciding with the completion of the initial phase of the Queensland Hepatitis B program), and the FNQ HBV clinical database (which is populated by the NOCS database and also draws data from local public and private laboratory and radiology services).

Inclusion/exclusion criteria is clear – only people that were alive, with confirmed CHB (2 sAg positive tests at least 6 months apart. Duplicates, patients with cleared infection, those that died or no longer lived in FNQ, and patients were lost to follow up (“unable to be identified despite contacting the clinic where the patients were initially diagnosed and last known to have had blood tests”) were excluded. Definitions of “engaged in care”, “eligible for treatment”, and cirrhosis are reasonable.

The indigenous Australian communities are especially vulnerable to infection with chronic hepatitis B, and also to disparities in delivery and access to standard of care. Therefore, it is important to share hepatitis B-related epidemiologic data for this population. While the prevalence of hepatitis B infection among these population was recently reported (Graham et al., Chronic hepatitis B prevalence in Australian Aboriginal and Torres Strait Islander people before and after implementing a universal vaccination program: a systematic review and meta-analysis Sexual Health, 2019, 16, 201–211), there is a paucity of publications reporting the uptake of hepatitis B care. Overall, this is a nice, brief description of the current chronic hepatitis B care continuum and HCC cases among the Aboriginal and Torres Strait Islanders of FNQ. However, in its current draft, it is not clear how the findings advance our knowledge/science of CHB, nor how these findings can be used to improve hepatitis B care. A few specific suggestions for the author’s consideration to strengthen this manuscript:

It is unclear what the aim of the study is. In the introduction lines 114-125 the authors talk about implementation of the national vaccination program and the aim of this study being to “evaluate the temporospatial epidemiology of HBV infection and clinical burden of disease in remote FNQ after the introduction of the vaccine”. But the results have not been clearly separated into pre-and post-vaccination program. The title is “Optimising the care of patients with chronic hepatitis B in remote, tropical Australia”. But discussion of interventions to optimize care have not been included. Furthermore, the link between their findings and how this can be used to optimize care has not been described and discussed. Clarification of the aim of this study would help.

Figure 2 clearly shows a decrease in prevalence after implementation of the vaccination program. But this finding is not referenced in the text nor legend – would help the reader if the change in prevalence is actually included in the text or legend.

The authors should consider restructuring their discussion to be more focused, based on whatever core message they are trying to convey.

6. PLOS authors have the option to publish the peer review history of their article (what does this mean?). If published, this will include your full peer review and any attached files.

Reviewer #1: No

Reviewer #2: No

---

## [Author Response · Author response to Decision Letter 0]

6 Aug 2020

Dear Professor Khudyakov,

We thank the Reviewers and the Editorial Staff for their constructive suggestions on how we might improve our manuscript. 

Thank you for the opportunity to respond to these suggestions. Please find below a point by point response to all the queries raised by the Editorial Staff and the Reviewers.

For clarity, we have presented the Editorial and Reviewers’ comments in blue and presented our responses in black.

If you require any further clarification, please do not hesitate to contact me.

Dr Josh Hanson (on behalf of all the authors)

Journal Requirements:

journals.plos.org/plosone/s/file?id=wjVg/PLOSOne_formatting_sample_main_body.pdf

and

journals.plos.org/plosone/s/file?id=ba62/PLOSOne_formatting_sample_title_authors_affiliations.pdf

Response: We apologise for the formatting errors in the original submission. We have reformatted the manuscript in accordance with the above instructions. We have also renamed the files in the resubmission as directed.

2. To comply with PLOS ONE submission guidelines, in your Methods section, please provide additional information regarding your statistical analyses.

In addition, please report your p-values to support your claims.

For more information on PLOS ONE's expectations for statistical reporting, please see https://journals.plos.org/plosone/s/submission-guidelines.#loc-statistical-reporting

Response: All the data presented in the manuscript are descriptive. There was limited analysis beyond determining medians and interquartile ranges. Accordingly, there are no “p values” to report. We have revised the statistics section in the methods to reflect this.

Response: The State of Queensland’s Public Health Act of 2005 precludes us from releasing these data into the public sphere. The data are collated using the State’s Notifiable diseases Conditions (NOCS) database and are bound by this act. However, the data are available from the Far North Queensland Human Research Ethics Committee (contact via email Cairns_Ethics@health.qld.gov.au) for researchers who meet the criteria for access to confidential data.

Please see the relevant legislation here:

(https://www.health.qld.gov.au/system-governance/legislation/specific/public-health-act#:~:text=The%20Public%20Health%20Act%202005,reducing%20risks%20to%20public%20health)

Response: We apologise again for this oversight. We have amended the manuscript as suggested in these instructions.

Reviewers' comments:

Reviewer's Responses to Questions

Comments to the Author

1. Is the manuscript technically sound, and do the data support the conclusions?

Reviewer #1: Partly

Reviewer #2: Yes

Response: We will address reviewer 1’s specific concerns below.

2. Has the statistical analysis been performed appropriately and rigorously?

Reviewer #1: Yes

Reviewer #2: N/A

Response: As the data presented are purely descriptive, no statistical analysis – beyond simple presentation of medians and interquartile ranges - has been performed.

3. Have the authors made all data underlying the findings in their manuscript fully available?

Reviewer #1: No

Reviewer #2: No

Response: As described above, The State of Queensland’s Public Health Act of 2005 precludes us from releasing these data into the public sphere. The data are collated using the State’s Notifiable diseases Conditions (NOCS) database and are bound by this act. However, the data are available from the Far North Queensland Human Research Ethics Committee (contact via email Cairns_Ethics@health.qld.gov.au) for researchers who meet the criteria for access to confidential data.

4. Is the manuscript presented in an intelligible fashion and written in standard English?

Reviewer #1: Yes

Reviewer #2: Yes

Response: We thank the reviewers for the time that they have taken to review our manuscript.

5. Review Comments to the Author

Reviewer #1: This descriptive manuscript reports the prevalence of chronic hepatitis B in remote Australia amongst aboriginal and Torres Strait Islander Australians. They report that 12% of individuals identified with hepatitis B in this region had chronic hepatitis B after introduction of universal vaccination. Of those engaged in care two thirds who qualified for antiviral therapy were receiving appropriate antiviral therapy. Low rates of hepatocellular carcinoma screening were also reported.

The article is important in that it demonstrates an inequity in access to healthcare within the robust Australian system. However the manuscript is quite lengthy, for the amount of data presented, particularly the discussion.

I would suggest the following modifications

1, The discussion is quite long for the data presented. The solutions proposed by the authors certainly deserve further attention but many of the suggestions are not really related to hepatitis B but rather related to improving general overall health (smoking etc.). The authors truncate the discussion some. As a minor point, I might change the title of the manuscript to something like”Epidemiology of hepatitis be in remote tropical Australia challenges and opportunities” or something similar.

Response: We agree with the reviewer and in the revised manuscript we have edited the text significantly, removing approximately 25% of the discussion (2607 words down to 1960 words). We have preferentially removed discussion around general Indigenous health as per the reviewer’s suggestion.

We have also revised the title of the manuscript as the reviewer has suggested. 

2. There are a fair number of individuals with hepatitis B were born after universal vaccination. A description of the prenatal care available in this region of Australia would be very helpful for the readers to gain better context of why infants may not be vaccinated.

Response: We agree with the reviewer and we have expanded this section to emphasise the failures (62 new diagnoses in patients born after the implementation of the vaccination programme) and successes (no cases in anyone born after 2014) of perinatal care. We have also provided a copy of the local management algorithm that has been employed recently to improve the antenatal and perinatal care of mothers living with hepatitis B and their babies.

3. To place the shortcomings of care of hepatitis B patients in this remote part of Australia, can the authors compare rates of liver cancer, appropriate initiation of antiviral therapy, and hepatocellular cancer screening from this region to other more developed parts of Australia.

Response: We have revised the text to address the reviewer’s concerns, although we have shifted the emphasis to highlight some of the successes of the local programme (no new infant diagnoses since 2014, and – by national standards - relatively high engagement in care and rates of antiviral prescription. We present national antiviral prescribing data from the most recent ASHM mapping report (lines 286-288) and we also expand the discussion about surveillance highlighting that before intervention the proportion of patients living with CHB who received HCC surveillance (only 10% - reference 34). We have altered the emphasis of our discussion about HCC to highlight the relatively infrequent diagnosis of HCC in the region (approximately 1 case per year). 

4. The authors note that hepatitis C screening has been quite successful in this region. Australia had a robust effort to eliminate hepatitis C. Are the same public health efforts being made for hepatitis B?

Response: We agree with the reviewer that Australia has made a robust effort to eliminate hepatitis C and that the approach to the management of hepatitis B has been less vigorous. There are several reasons for this. First, hepatitis C can be cured with a relatively short course of therapy which hepatitis B cannot (a point we make in lines 264-266). Also, hepatitis B has a higher prevalence in remote locations, in Indigenous populations and in people of a non-English speaking background – all factors which make them harder to reach and deliver optimal care. We have expanded our discussion to address these points (lines 268-270)

5. While the HCC screening rates are disappointing, the authors should note that their screening rates are not different than other parts of the world. Can the authors also compared to more developed regions of Australia.

Response: The reviewer makes a good point. We have specifically highlighted the reported figure of 10% from a study performed in metropolitan Melbourne. There are also several references that highlight the low uptake and limitations of HCC screening (lines 298-300 and references 31-34).

Reviewer #2: In Australia, prevalence among its Indigenous Aboriginal and Torres Strait Islander people (Indigenous Australians) is disproportionately high, at 4.0% compared to 0.95% in the general population. Subsequently liver-related deaths in Australia re 3.7 times more common in Indigenous than in non-Indigenous Australians. Many Indigenous Australians live in remote locations, where access to health care is limited and where other health conditions compete for finite resources. This study evaluates the temporospatial epidemiology of HBV infection and clinical burden of disease in remote Far North Queensland (where the majority of the population identify as Aboriginal and/or Torres Strait Islander people) after the introduction of the hepatitis B vaccine. It was hoped that these data might define the scale of the local problem and inform future cost effective strategies to reduce CHB-related morbidity and mortality in the region.

This a retrospective study using data from the hepatitis B disease registry in the Queensland notifiable conditions system (NOCS) database, from January 1, 1990 to December 31, 2019 (coinciding with the completion of the initial phase of the Queensland Hepatitis B program), and the FNQ HBV clinical database (which is populated by the NOCS database and also draws data from local public and private laboratory and radiology services).

Inclusion/exclusion criteria is clear – only people that were alive, with confirmed CHB (2 sAg positive tests at least 6 months apart. Duplicates, patients with cleared infection, those that died or no longer lived in FNQ, and patients were lost to follow up (“unable to be identified despite contacting the clinic where the patients were initially diagnosed and last known to have had blood tests”) were excluded. Definitions of “engaged in care”, “eligible for treatment”, and cirrhosis are reasonable.

The indigenous Australian communities are especially vulnerable to infection with chronic hepatitis B, and also to disparities in delivery and access to standard of care. Therefore, it is important to share hepatitis B-related epidemiologic data for this population. While the prevalence of hepatitis B infection among these population was recently reported (Graham et al., Chronic hepatitis B prevalence in Australian Aboriginal and Torres Strait Islander people before and after implementing a universal vaccination program: a systematic review and meta-analysis Sexual Health, 2019, 16, 201–211), there is a paucity of publications reporting the uptake of hepatitis B care. Overall, this is a nice, brief description of the current chronic hepatitis B care continuum and HCC cases among the Aboriginal and Torres Strait Islanders of FNQ. However, in its current draft, it is not clear how the findings advance our knowledge/science of CHB, nor how these findings can be used to improve hepatitis B care. A few specific suggestions for the author’s consideration to strengthen this manuscript:

It is unclear what the aim of the study is. In the introduction lines 114-125 the authors talk about implementation of the national vaccination program and the aim of this study being to “evaluate the temporospatial epidemiology of HBV infection and clinical burden of disease in remote FNQ after the introduction of the vaccine”. But the results have not been clearly separated into pre-and post-vaccination program. The title is “Optimising the care of patients with chronic hepatitis B in remote, tropical Australia”. But discussion of interventions to optimize care have not been included. Furthermore, the link between their findings and how this can be used to optimize care has not been described and discussed. Clarification of the aim of this study would help.

Response: We agree with the reviewer. The original manuscript was unfocussed and lacked a clear narrative. We have revised the manuscript to highlight the aim of the study more succinctly

“This study was performed to evaluate the temporospatial epidemiology of the disease in remote FNQ, to define the present burden of HBV infection. The study also assessed local clinicians’ adherence to components of HBV care that are currently recommended in National Guidelines” (lines 122-126)

We address the temporospatial epidemiology in Figures 2 (time) and 3 (space) and the discussion (lines 237-257).

We address the burden of disease in the results in Table 1 and in lines 205-219 and in the discussion. 

We address local clinicians’ adherence to care in table 1 and the discussion (lines 277-293).

Figure 2 clearly shows a decrease in prevalence after implementation of the vaccination program. But this finding is not referenced in the text nor legend – would help the reader if the change in prevalence is actually included in the text or legend.

Response: We thank the reviewer for highlighting this point. We have amended the legend to figure 2 as suggested. 

However, it is not possible to determine the true local prevalence of HBV from these data as individuals who are still children (who represent a significant proportion of the people born after 1985) will not necessarily have repeated blood testing performed as is the case with adults. These data do not represent a serosurvey, merely the cases that have been identified (a limitation that we highlight in lines 342-345). Indeed, even the baseline prevalence of >20% was based on relatively small serosurveys in the 1980s. 

Despite this, the figure, is, we believe illustrative of the expected – and biologically plausible - effect of vaccination and emphasises that the most efficient and cost-effective way to reduce the burden of hepatitis B related complications is to prevent the infection in the first place.

The authors should consider restructuring their discussion to be more focused, based on whatever core message they are trying to convey.

Response: We agree with the reviewer and have excised much of the less focussed discussion (approximately 25% of the original submission). 

We now focus purely on the burden of HBV infection and strategies to reduce its incidence (vaccination and antenatal care) and its complications (employment of Aboriginal and Torres Strait Islander health workers, increased anti-viral prescription, de-emphasis of HCC surveillance). 

While much of the other discussion about general Indigenous Health is important for clinicians working in the region, we agree that it was beyond the scope of the article and distracted the reader from the main messages

---

## [Editor Report · Decision Letter 1]

24 Aug 2020

Chronic hepatitis B in remote, tropical Australia; successes and challenges

PONE-D-20-17114R1

Dear Dr. Hanson,

We’re pleased to inform you that your manuscript has been judged scientifically suitable for publication and will be formally accepted for publication once it meets all outstanding technical requirements.

Kind regards,

Yury E Khudyakov, PhD

Academic Editor

PLOS ONE
---

## [Editor Report · Acceptance letter]

26 Aug 2020

PONE-D-20-17114R1 

Chronic hepatitis B in remote, tropical Australia; successes and challenges 

Dear Dr. Hanson:

I'm pleased to inform you that your manuscript has been deemed suitable for publication in PLOS ONE. Congratulations! Your manuscript is now with our production department. 

Kind regards, 

on behalf of

Dr. Yury E Khudyakov 

Academic Editor

PLOS ONE